# Status and future prospect of deregistered woodland key habitats in Northwestern Sweden

**Ulrika Ervander**[1,2]*, **Willian T. A. F. Silva**[2,3], **Karin C. Harding**[2], **Micael Jonsson**[4]

**1** Department of Earth Sciences, University of Gothenburg, Gothenburg, Sweden, **2** Department of Biology and Environmental Sciences, University of Gothenburg, Gothenburg, Sweden, **3** Department of Aquatic Resources, Swedish University of Agricultural Sciences, Lysekil, Sweden, **4** Department of Ecology, Environment and Geocience, Umeå University, Umeå, Sweden

* ulrika.ervander@gu.se

## Abstract

Extensive primary boreal forests within Europe are primarily located in Fennoscandia and northwestern Russia. These forests host numerous endemic and red-listed species but are rapidly being exploited and transformed to production forests that lack the habitat characteristics that are required for sustained biodiversity. Over the past 30 years, certain highly valuable areas within Swedish forests have been designated and registered as "woodland key habitats" (WKH) to be safeguarded from clear-cutting. However, despite their high conservation values, WKH lack proper legislative protection. Recently, many WKH were deregistered and thereby lost their potential protection against clear-cutting, jeopardizing biodiversity values in these forests. Moreover, the former way of classifying WKH has been criticized for being too lenient, making conventional forestry difficult. To assess the leniency of WKH registration and effect of WKH deregistration, we conducted a field inventory of WKH following a new inventory method proposed by the Swedish Forest Agency, featuring more stringent criteria for classification of WKH in north-western Sweden. The inventory was conducted in 9 still registered and 9 recently deregistered WKH to assess their conservation values. Our inventory results show that all 18 areas reach the criteria for WKH with the new method, despite higher, more stringent thresholds for conservation values. Hence, formerly registered WKH were not deregistered due to lower values. Moreover, analysis of recent harvest actions within deregistered WKH in Sweden showed that almost 1,200 hectares (~ 2%) of these areas were clear-cut or reported for clear-cutting 1–7 years post deregistration. As such, our results indicate that WKH contain high values, even using more stringent classification criteria, but also that deregistration of WKH does not consider these values and increases the risk of losing them to forest management. Given past and current declines in forest biodiversity, this is concerning, as conservation of areas containing high conservation values are needed, in order to preserve biodiversity in boreal forests.

**Data availability statement:** "The field data collected for this study are available as Supporting Information. Table S1 includes all information on plots within each area, and Table S2 includes all collected data in field. The data used for geographical analysis in GIS on harvest and harvest reports are publicly available from Skogsstyrelsen ((2020) Map of woodland key habitats from Swedish Forest Agency)) (https://www.skogsstyrelsen.se/sjalvservice/karttjanster) and (2023) (https://www.skogsstyrelsen.se/sjalvservice/karttjanster/geodatatjanster/). The data are available directly from the website, and for this study the authors downloaded shapefiles within the date interval described in the paper. Data on deregistered key habitats were gathered by the forest group in Kronoberg within the Swedish Society for Nature Conservation (Skogsgruppen Kronobergs Naturskyddsförening (2019)). These data are available in the Supporting Information files (skogsgruppen_data_2019.shp).".

**Funding:** The author(s) received no specific funding for this work.

**Competing interests:** the authors have declared that no competing interests exist.

## 1. Introduction

Primary forests and intact forest landscapes worldwide play a crucial role in maintaining ecological diversity and mitigating the impacts of global warming [1]. Despite growing awareness of their importance, primary forests continue to decline and become increasingly fragmented at an alarming rate, due to intensive forest management [2,3]. Approximately 44% of the intact forest landscapes globally consist of boreal forests [1]. In Europe, large intact boreal forests are now restricted to northwestern Russia and Fennoscandia [4,5]. In northwestern Sweden, there are still old-growth forests that have never been clear-cut, which gives these forests a unique status in the European Union [4]. This green belt of old-growth forest in the Swedish northwest sustains rich biodiversity and holds significant conservation values, but despite their unique ecological importance, they are constantly diminishing due to high demands for wood products and subsequent itensified forest management [3,6]. The rapidly altered forest landscape in boreal Sweden has resulted in a substantial loss of both habitat area and quality of habitat for red-listed species and 'signal species' [7]. Signal species are indicators for forests of old age that contain high conservation values, including old trees, large amounts of dead wood, and high spatial heterogeneity [8]. Out of the 4,127 species listed on the Swedish red list, 2,131 are associated with this forest biotope [7]. Notably, 75% of these forest species are red-listed because of the diminishing habitat and reduced habitat quality due to clear-cutting of old-growth forests [7]. Such conversion of continuous primary forests and old-growth forests into even-aged monocultures void of dead wood is recognized as having the most pronounced negative impact on red-listed species in the Nordic countries and contributes significantly to the ongoing loss of biodiversity globally [7,9,10].

In 1990, the Swedish Forest Agency (SFA) initiated the identification of Woodland Key Habitats (WKH) to pinpoint the most valuable areas within Swedish forests [11,12], not as a formal protection mechanism [13]. Its protective role has emerged indirectly through certification standards such as FSC, which prohibit harvesting in WKHs, and PEFC, which requires protection of WKH of at least 5% [14,15]. However, WKHs lack features essential for long-term conservation, such as buffer zones and larger continuous green infrastructure including these hotspots [16,17]. Currently, many WKHs are relatively small, with an average area of 4.6 hectares [18], often relics in a fragmented forest landscape that serve as refuges for threatened species and thereby can host source populations for future recolonization efforts [19,20]. The small size of WKH is a widespread problem throughout Europe, and this limited size leads to strong edge effects that are likely to negatively impact their ecological quality [21,22]. A WKH is a forest area that, based on an overall assessment of its structural features, species composition, historical use, and current physical environment, is considered to be of critical importance for forest biodiversity. WKH are defined as habitats capable of supporting red-listed species [12]. On average, more than 20 red-listed and signal species can be found in a single WKH [11]. It has been proposed that special attention should be given to forests adjacent to WKH, to enhance the long-term spread and survival of red-listed species [11].

An assessment of a recently developed method for classifying WKH by the SFA revealed that over 55,000 hectares of WKH were subjected to clear-cutting between 2000 and 2018 [23]. The logging of valuable WKH occurs despite the fact that most big forest companies in Sweden are certified according to the standards of the FSC, with criteria that do not allow harvesting of WKH [14]. However, even if a WKH classification signifies a forest's conservation values based on inventory results, it does not confer formal legal protection [20].

The WKH inventory method relies on a visual inventory, wherein the surveyor visits a forest identified as a likely WKH to document the presence and frequency of structural attributes and species associated with old-growth forests. The forest history, age, canopy layers and elements of high conservation value are documented, and these data serve as the basis for the classification of the area [12]. In the inventory method, SFA has developed checklists for northern Sweden, with the aim to promptly identify areas with high conservation value, using a straightforward and practical method. These checklists were primarily developed for surveyors lacking biological training or expertise in classifying a WKH but have generally not been consistently and systematically applied [13]. Further, applying the same method for WKH inventories across the whole country can be problematic, as the forest landscapes, which span 1,570 km from north to south, shows substantial regional variations. For example, a major part of the old-growth forests in northwestern Sweden is within or close to the mountain region and therefore differ substantially from forests in other regions, in terms of climate (lower temperatures), productivity (lower light incidence), and accessibility (difficult terrain) [24]. Consequently, these forests have a history of less extensive exploitation, resulting in a significantly higher proportion of natural old-growth forests compared to elsewhere in Sweden, and the formally protected areas as well as the voluntary set-aside areas are larger. In these forests, registered WKH are also more common, and it is highly probable that there are numerous areas still unidentified as WKH [13].

The inventory method for WKH has remained relatively unchanged since the start of the WKH identification survey in 1990, with the same method applied across the country. However, in 2017, the SFA decided to halt the WKH inventory work in northwestern Sweden [12], despite it having the highest density of unprotected old forests [24]. During this period, it was also discovered, through a comparison of new and old maps of registered WKH, done by the Swedish Society for Nature Conservation in Kronoberg 2019, that many WKH were deregistered and in some cases the deregistration was followed by forest harvest. In 2021, the SFA announced that they will stop the WKH inventories for the entire country, shifting the responsibility for identifying these areas to the forest owner. The decision to halt WKH inventories in 2017 was driven by the need to develop a new method that could address challenges in assessing and delimiting WKH in a more effective way, especially in regions with extensive areas of old forests [25]. The inventory suspension continued until January 2018, followed by the new inventory method being evaluated throughout 2018 and then published in 2019. The new method is intended to be more systematic and with higher thresholds for an area to qualify as a WKH in northwestern Sweden [12]. The classification of an area is determined by specific criteria outlined in a checklist, which is applied within each area. The result obtained from the checklists should be used as a strong guidance in the classification of a forest; however, the classification of a WKH also involves a comprehensive assessment of both the forest and the surrounding landscape [12]. In this study, we evaluate the performance of the newly developed survey method for WKH in northwestern Sweden and compare the conservation value of currently registered WKH with that of recently deregistered ones. Our three main aims are to:

1. Compare registered and deregistered WKH through field inventories based on the following criteria in the new method for WKH; number of conservation trees, number of old trees, number of deadwood objects, and number of signal and red-listed species.

2. Quantify the potential consequences of changes in inventory methodology and the ongoing deregistration of these valuable forest areas by analyzing harvest records and pre-harvest assessments within deregistered WKH.

3. Illustrate the potential long-term impact of deregistration of WKH on forest conservation values and biodiversity.

## 2. Materials and methods

### 2.1 Study area

The fieldwork was carried out in the boreal forests in northwestern Sweden close to the town of Sorsele in Västerbotten county (65°31'40"N, 17°31'49"E) during September and October 2019. Within the study area, covering a 20-km radius around Sorsele (Fig 1A), all registered and recently deregistered WKH, owned by the state-owned forest company Sveaskog AB, were identified on maps obtained from the SFA. Out of the 96 identified areas initially classified as WKH using the old method, 41 of them have been deregistered, and thus lost their status as WKH between 2015–2018. Among these areas, 9 registered and 9 deregistered WKHs were randomly selected for the study. Only forests managed by the state-owned forest company, Sveaskog AB, were included, to minimize variations arising from company-specific inventory methods. The map of deregistered WKHs included in this study is provided as supporting information and is based on work by the Forest Group of the Swedish Society for Nature Conservation in Kronoberg 2019.

The software QGIS (3.16.15 Hannover) was used to generate a grid of 200 m x 200 m squares on the map of the study area. The number of sample sites per forest area was determined according to the SFA's guidelines for the new developed inventory method. For objects <0.5 ha, one sample plot was placed in a representative location. For objects between 0.5 and 10 ha, 2–5 plots were placed either on a grid or in representative areas. For objects >10 ha, one plot was placed per 4 ha using a grid-based layout (Fig 1). Coordinates and number of sample sites are presented in supporting information. The deregistered WKH included in this study underwent deregistration between 2015 and 2018, before the SFA stopped identifying and registering WKHs in 2021. The classification of the deregistered WKH at the time of the field survey is presented in Table 1. Forest D2 (Fig 1, Table 1) is classified as a production forest and the remaining eight deregistered forests were re-labeled as "non-exchangeable promises" which is a category of voluntary set-aside area with a higher protection from exploitation through forestry than a regular set-aside area [26] and relies on an agreement between Sveaskog and the county administrative board [27]. However, during 2023, these eight "non-exchangeable promises" areas were reclassified once again and are now included in an area for long-term protection. This area also covers an additional 2,149 hectares of deregistered WKH, out of the more than 61,000 hectares within the map of deregistered WKH included in this study [28].

### 2.2 Field survey

Each forest area was surveyed according to the field manual for the new method for identifying WKH in northwestern Sweden [12]. The field manual includes a checklist of parameters, adjusted according to the type of forest (i.e., spruce-dominated, pine-dominated, etc.). For each forest area there is a number of sample sites and at each sample site an inventory is done following the checklist. Our field survey was conducted during snow-free conditions to ensure full visibility of the forest floor. The sample sites are circular plots with a radius of 25 meters, corresponding to an area of approximately 0.2 hectares. All recorded objects within each plot were thus multiplied by 5 to estimate the number of objects per hectare. The checklist has a list of criteria, and a forest is defined as a likely WKH if a certain number of criteria are met. The criteria on the checklist differ depending on forest type but all checklists include the following parameters: conservation trees of different species per hectare, lying and standing deadwood of different species per hectare, continuity of tree age, continuity of dead wood (i.e., number of deadwood objects > 10 cm in diameter per hectare), number of old trees per hectare, presence of red-listed and signal species and other descriptors specific to the surveyed forest type [23]. The term "conservation tree" is defined in the method by the SFA as a tree with significant importance for biodiversity, such as trees older than 250 years, trees with hanging lichens, or those with suitable cavities for cavity-nesting birds. Trees that are likely to in the future develop properties of great importance for biodiversity are also included within the term "conservation tree". Red-listed and signal species do not need to be found within the sample site to be included but are added to the inventory list if seen in between sample sites within the forest area that is surveyed. All signal and red-listed species are included in the overall assessment of the key biotope [23].

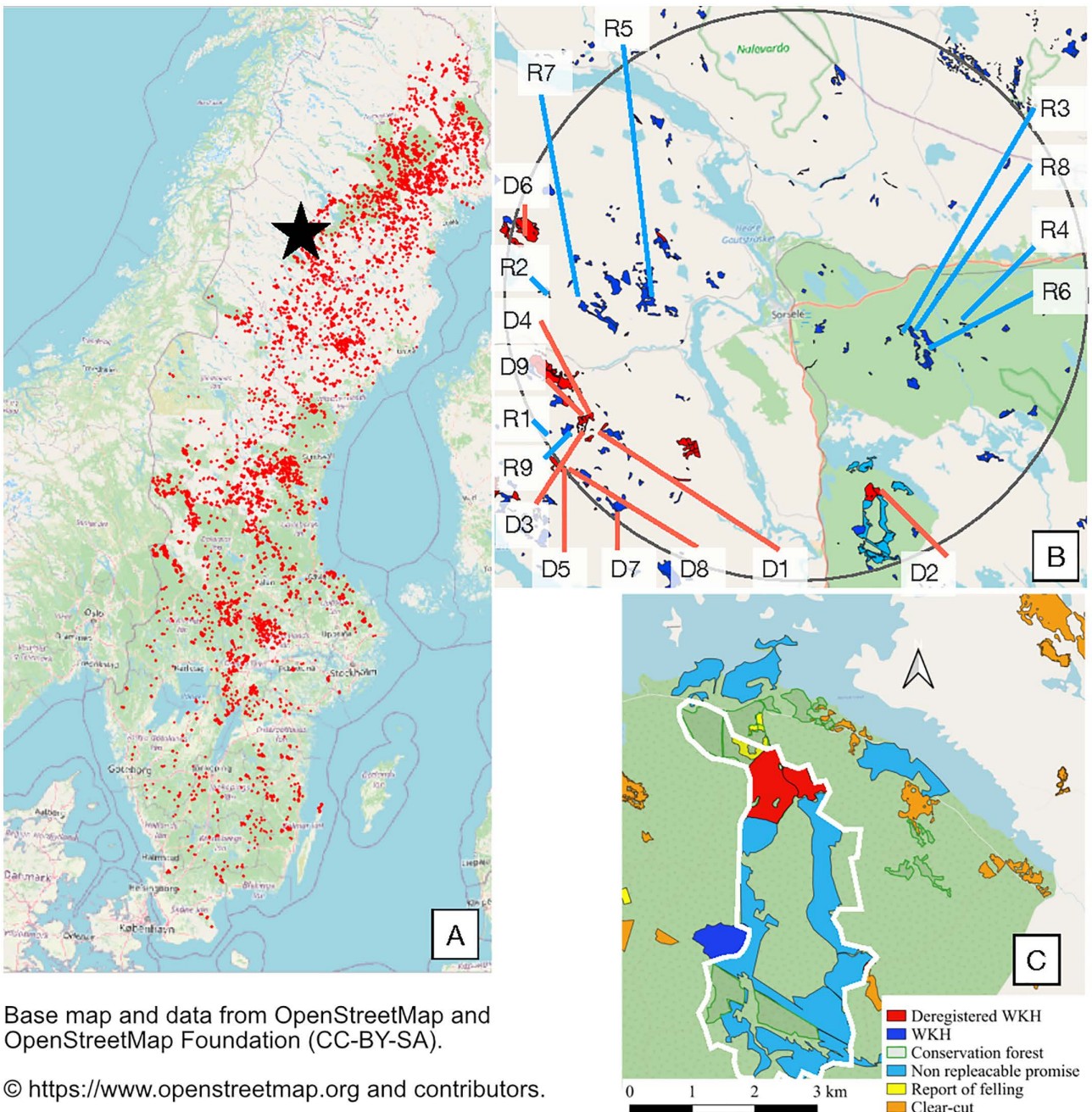

Base map and data from OpenStreetMap and
OpenStreetMap Foundation (CC-BY-SA).

© https://www.openstreetmap.org and contributors.

**Fig 1. Map of study area.** Map of WKH in Sweden. The star in map A denotes the location of the study area in northwestern Sweden, with red dots illustrating the distribution of all deregistered WKH included in the map used in this study. Map B is a close-up of the study area, where all WKH are indicated in blue and deregistered WKH are indicated with red. All surveyed areas are marked with a label. Map C presents deregistered WKH D2 (now classified as production forest), in a landscape perspective. Boundaries of forests of high conservation value according to the County Administrative Board are delineated by a white line. [34–36-Base map from OpenStreetMap https://www.openstreetmap.org/].

**Table 1. Summary of results from the field survey.**

| ID | Criteria track 1 | Criteria track 2 | deregistered (year) | Class-At time of survey | Survey result | Red-listed and signal species (classification from red list) |
|---|---|---|---|---|---|---|
| R1 | 6/6 | 18/22 | | WKH | WKH | *Phlebia mellea* (VU), *Dryocopus martius* (NT), *Lobaria pulmonaria* (NT), *Lobaria scrobiculata* (NT), *Alectoria sarmentosa* (NT), *Phellinus chrysoloma* (NT), *Nephroma bellum* (LC) |
| R2 | 6/6 | 15/22 | | WKH | WKH | *Phellinus chrysoloma* (NT), *Alectoria sarmentosa* (NT), *Fomitopsis rosea* (NT) |
| R3 | 5/6 | 16/22 | | WKH | WKH | *Haploporus odorus* (VU), *Diplomitoporus lenis* (VU), *Antrodia albobrunnea* (VU), *Alectoria sarmentosa* (NT), *Lobaria pulmonaria* (NT), *Dryocopus martius* (NT), *Phellinus chrysoloma* (NT), *Lobaria scrobiculatus* (NT), *Nephroma bellum* (LC), *Nephroma resupinatum* (LC), *Icmadophila ericetorum* (LC) |
| R4 | 6/6 | 15/22 | | WKH | WKH | *Alectoria sarmentosa* (NT), *Phellinus chrysoloma* (NT), *Picoides tridactylus* (NT), *Pseudographis pinicola* (NT) |
| R5 | 6/6 | 12/22 | | WKH | WKH | *Alectoria sarmentosa* (NT), *Phellinus chrysoloma* (NT), *Lobaria scrobiculata* (NT), *Icmadophila ericetorum* (LC) |
| R6 | 5/6 | 14/22 | | WKH | WKH | *Alectoria sarmentosa* (NT), *Dryocopus martius* (NT), *Cystostereum murrayi* (NT), *Phellinus chrysoloma (NT)*, *Lobaria scrobiculatus* (NT), *Nephroma resupinatum* (LC) |
| R7 | 6/6 | 16/22 | | WKH | WKH | *Phellinus chrysoloma* (NT), *Alectoria sarmentosa* (NT), *Perisoreus infaustus* (LC), *Matteuccia struthiopteris* (LC) |
| R8 | 6/6 | 18/22 | | WKH | WKH | *Fomitopsis rosea* (NT), *Leptoporus mollis* (NT), *Picoides tridactylus* (NT), *Alectoria sarmentosa* (NT), *Pseudographis pinicola* (NT), *Phellinus chrysoloma* (NT), *Perisoreus infaustus* (LC), *Nephroma bellum* (LC) |
| R9 | 6/6 | 16/22 | | WKH | WKH | *Cystostereum murrayi* (NT), *Alectoria sarmentosa* (NT), *Phellinus chrysoloma* (NT), *Onnia leporina* (NT), *Perisoreus infaustus* (LC), *Icmadophila ericetorum* (LC), *Aconitum lycoctonum* (LC), *Icmadophila ericetorum* (LC), *Nephroma resupinatum* (LC) |
| D1 | 6/6 | 16/22 | 2015 | promise | WKH | *Alectoria sarmentosa* (NT), *Phellinus chrysoloma* (NT), *Icmadophila ericetorum* (LC) |
| D2 | 2/6 | 12/22 | 2017 | production | WKH | *Haploporus odorus* (VU), *Skeletocutis odora* (VU), *Phellinidium ferrugineofuscum* (NT), *Carbonicola anthracophila* (NT), *Dryocopus martius* (NT), *Pseudographis pinicola* (NT), *Alectoria sarmentosa* (NT), *Phellopilus nigrolimitatus* (NT), *Onnia leporina* (NT), *Nephroma resupinatum* (LC) |
| D3 | 6/6 | 18/22 | 2015 | promise | WKH | *Poecile cinctus* (VU), *Alectoria sarmentosa* (NT), *Phellinus chrysoloma* (NT), *Onnia leporina* (NT), *Picoides tridactylus* (NT), *Nephroma bellum* (LC), *Nephroma resupinatum* (LC), *Meruliopsis taxicola* (LC) |
| D4 | 6/6 | 18/22 | 2015 | promise | WKH | *Skeletocutis odora* (VU), *Fomitopsis rosea* (NT), *Phellinus chrysoloma* (NT), *Alectoria sarmentosa* (NT), *Phellinidium ferrugineofuscum* (NT), *Leptoporus mollis* (NT), *Perisoreus infaustus* (LC), *Icmadophila ericetorum* (LC) |
| D5 | 6/6 | 15/22 | 2015 | promise | WKH | *Alectoria sarmentosa* (NT), *Phellinidium ferrugineofuscum* (NT), *Fomitopsis rosea* (NT), *Onnia leporina* (NT), *Phellinus chrysoloma* (NT), *Phellopilus nigrolimitatus* (NT), *Icmadophila ericetorum* (LC) |
| D6 | 5/6 | 18/22 | 2015 | promise | WKH | *Antrodia albobrunnea* (VU), *Diplomitoporus lenis* (VU), *Skeletocutis odora* (VU), *Phlebia mellea* (VU), *Laurilia sulcata* (VU), *Onnia leporina* (NT), *Alectoria sarmentosa* (NT), *Phellinus chrysoloma* (NT), *Pseudographis pinicola* (NT), *Cystostereum murrayi* (NT), *Leptoporus mollis* (NT), *Phellinidium ferrugineofuscum* (NT), *Lobaria pulmonaria* (NT), *Fomitopsis rosea* (NT), *Phellopilus nigrolimitatus* (NT), *Picoides tridactylus* (NT), *Hypogymnia bitteri* (NT), *Icmadophila ericetorum* (LC), *Meruliopsis taxicola* (LC), *Perisoreus infaustus* (LC), *Nephroma bellum* (LC) |
| D7 | 6/6 | 13/22 | 2015 | promise | WKH | *Phellinus ferrugineofuscum* (NT), *Alectoria sarmentosa* (NT), *Onnia leporina* (NT), *Fomitopsis rosea* (NT), *Nephroma bellum* (LC) |
| D8 | 6/6 | 16/22 | 2015 | promise | WKH | *Laurilia sulcata* (VU), *Phellinidium ferrugineofuscum* (NT), *Fomitopsis rosea* (NT), *Cystostereum murrayi* (NT), *Phellinus chrysoloma* (NT), *Phellopilus nigrolimitatus* (NT), *Alectoria sarmentosa* (NT), *Picoides tridactylus* (NT), *Icmadophila ericetorum* (LC), *Perisoreus infaustus* (LC) |
| D9 | 6/6 | 15/22 | 2015 | promise | WKH | *Alectoria sarmentosa* (NT), *Phellinus chrysoloma* (NT), *Pseudographis pinicola* (NT), *Nephroma bellum* (LC) |

Survey result of all 18 forests randomly selected from the study area. Criteria track 1 – forest likely a WKH if more than 5 criteria are fulfilled. Criteria track 2 – forest likely a WKH if more than eight or nine criteria are fulfilled (depending on the size and surrounding). Results of all findings of different red-listed and signal species from field survey. Abbreviations of conservation status from the red list is added after each species. Critically (CR), endangered (EN), vulnerable (VU), near threatened (NT) and least concern (LC) [7].

Within the checklist there are two different routes to reach the WKH classification status for a forest area (Track 1 and Track 2): in Track 1, a few high-value criteria suffice, while, in Track 2, a longer list of lower-value criteria is required. Track 1 and Track 2 are closely related but can be assessed independently. Track 1 includes key criteria that define forests with high conservation value; if enough are met, the area is likely classified as a WKH. Each criterion in Track 1 generally carries more weight than those in Track 2, which includes supporting factors that strengthen the likelihood of WKH classification. For example, in Track 1, having 30 conservation spruce trees per hectare is adequate to reach one of the six criteria, whereas in Track 2, 20 conservation spruce trees per hectare is required to reach one of the 22 criteria. In northwestern Sweden, a surveyed forest is designated as a WKH if it satisfies five out of the six criteria in Track 1. The number of criteria in Track 2 that must be fulfilled depends on the location as well as size of the forest: forests smaller than 20 hectares need to fulfill at least nine criteria, whereas larger forests need to fulfill at least eight criteria. All checklist and full description of the method used for the survey in this study are included in the report by the SFA [23]. The classification of a WKH involves a comprehensive assessment of both the forest stand and the surrounding landscape. In addition to quantifying specific parameters, the survey should also document signs of human disturbance and other features that may influence the forest's conservation value. Besides being more stringent, the new method also favors large areas and areas in proximity to other WKH to create a green infrastructure. According to the new method, a so-called "50/50 rule" can be applied in a continuous forest landscape characterized by high conservation values, especially when borders are hard to delimit [12]. This rule dictates that if 50% of the area reaches the criteria for WKH standard, the entire area should be designated as a WKH.

## 2.3 Statistical and geographical analyses

To assess differences between registered and deregistered WKHs, we focused on four key inventory parameters present in all checklists: (1) number of conservation trees, (2) number of old trees, (3) number of deadwood objects, and (4) number of signal and red-listed species.

For parameters 1–3, which were measured across multiple plots within each forest area, we fitted generalized linear mixed models (GLMMs) with a negative binomial error distribution using the glmmPQL function from the MASS package in R [29]. Registration status (registered vs. deregistered) was included as a fixed effect, and forest area as a random effect to account for the nested sampling structure (plots within forest areas). Initially, we fitted Poisson GLMMs using the glmer function in the lme4 package [30]. However, overdispersion (where variance exceeds the mean, violating Poisson assumptions) was detected based on Pearson residuals [31] (dispersion ratio > 2), prompting the use of a negative binomial distribution. For parameter 4 (number of red-listed signal species), only one number per forest was collected, so we fitted a negative binomial generalized linear model (GLM) with registration status as the predictor. A Poisson GLM was initially tested, but again overdispersion was evident (dispersion ratio > 2), warranting the negative binomial model.

Model diagnostics, including Q–Q plots and residual histograms were used to assess model fit for all parameters. All statistical analyses were conducted using the software R [32] and R-studio Version12.0 + 353 [33]. To study the frequency of harvesting in these areas, QGIS was used to analyze maps of conducted harvesting from the Swedish forest agency together with maps of deregistered WKHs. Geographical analysis was performed using QGIS (3.16.15 – Hannover) and the geoprocessing tool intersection. Geographical data of harvesting operations carried out within deregistered key habitats after the deregistration date were combined, and the intersecting area was calculated. The data of harvesting operations from SFA span from 2010-01-01 to 2023-03-02, but only data of harvesting operations from dates after deregistration of key habitats included in this study were analyzed [34].

## 3. Results

### 3.1 Testing the new inventory method

All surveyed forests fulfilled the criteria for WKH according to the new method for WKH classification in northwestern Sweden (Table 1). Forest D2 (located on Abmoberget) did not fulfill the criteria in both tracks, but still met the criteria in Track 2, which is sufficient for WKH classification (Table 1).

Forest D2 is part of a green infrastructure from a landscape perspective (Fig 1). In this case, the area of classified WKH in the surrounding forest landscape covers over 450 hectares and the two deregistered forests cover together less than 1/4th of the size of the surrounding WKH. The 50/50-method thus strengthens the assessment to classify this forest as a WKH.

### 3.2 Conservation values in registered and deregistered WKH.

A majority of the surveyed forests displayed no clear indications of human disturbance and appeared never to have undergone large-scale exploitation. The biodiversity and conservation value of all forests were notably high, all having more than 100 conservation trees per hectare and many red-listed species. The forests were old, dominated by spruce, with a high amount of dead wood objects and many old living trees with an age over 250 years and draped with pendulous lichens. The number of red-listed and signal species in the surveyed forests varied from 3 to 21 with an average of 7.4 species per WKH (Table 1). The largest deregistered WKH (D6) had the highest number of observed red-listed species, 18 distinct species, of which 5 classified as vulnerable (VU). In the deregistered WKH, D2, currently classified as production forest, nine different red-listed species were identified, including 2 VU species.

### 3.3  Comparison between findings in studied WKH and established criteria in new inventory method

All surveyed forests exceeded the set criteria of a minimum of 20 conservation trees per hectare for Track 1. Additionally, the number of spruce conservation trees was above the criteria in all forests, with at least 30 conservation trees per hectare (Fig 2). The negative binomial GLMM indicated a positive but marginally non-significant effect of registration status on the number of conservation trees per hectare ($p=0.086$), suggesting that registered WKHs tend to have more conservation trees than deregistered ones, although the difference was not statistically significant. The model included forest area as a random effect to account for variability among sites. Model diagnostics, including Q–Q plots and Pearson residual histograms, supported the adequacy of model fit and confirmed that overdispersion was appropriately handled by the negative binomial structure. No significant difference was found in the number of distinct signal and red-listed species between registered and deregistered forests ($p=0.372$; Fig 3). The total number of deadwood objects exceeded the set criteria of a minimum of 60 dead trees per hectare in Track 1 for all areas, except for WKH D2, which had a relatively low number of deadwood objects per hectare (Fig 4).

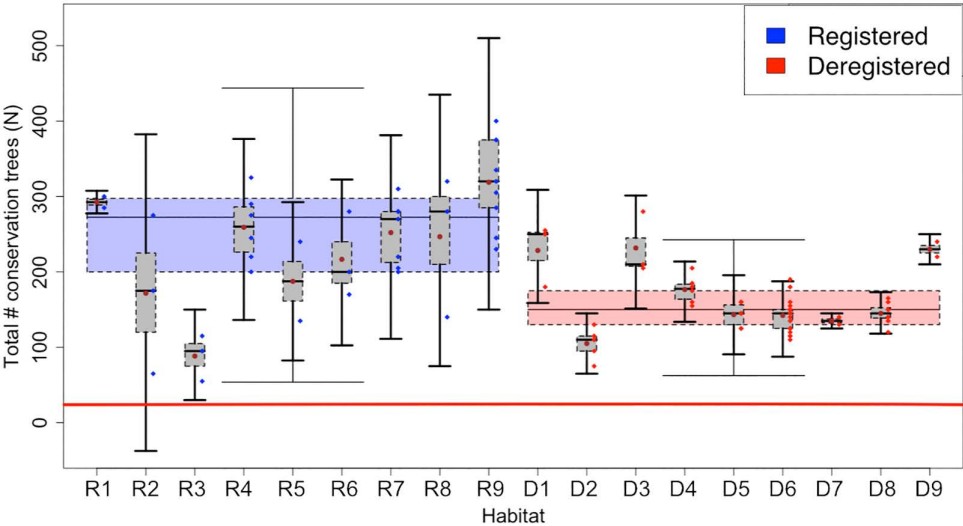

**Fig 2.  Conservation trees per hectare.** Mean number (± 1 SD), of conservation trees per hectare within each forest. The small box plots show the variation within each forest and the large box plots show the variation within the categories registered WKH (R) and deregistered WKH **(D)**. Red line shows criteria for track 1 by the new inventory method, i.e., 20 conservation trees per hectare. It can be noted that all forests are well above the limit of 20 trees per hectare set as a criterium for WKH.

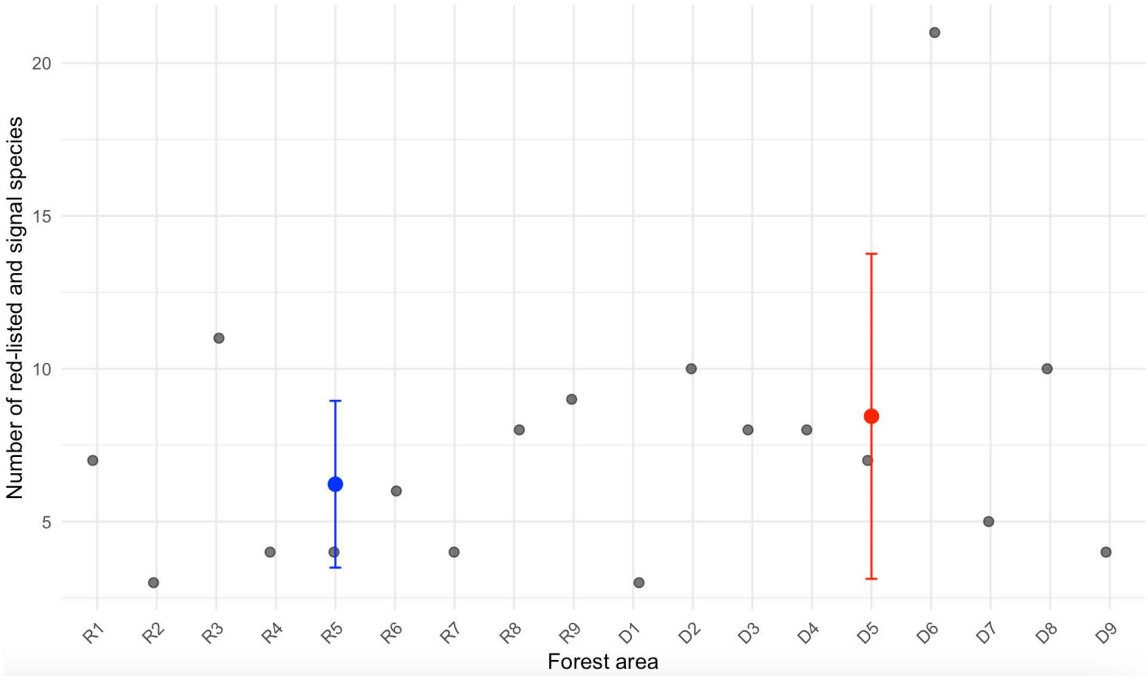

**Fig 3. Number of red-listed and signal species.** Mean number (± 1 SD), of the number of red-listed and signal species observed during survey within each forest. The box plots show the variation within the categories registered WKH (R1-9) and deregistered WKH (D1-9).

Both the total number of deadwood objects and the number of living trees of >250 years of age in the studied forests were much higher than the criteria in the new inventory method. The average number of deadwood objects (both standing and lying) across all areas was 192 per hectare, which is well above the criteria of 50/hectare in Track 1 and 60/hectare in Track 2 (Fig 4). The average number of deadwood objects per site exceeded the threshold in all areas except D2, where the mean density was 62 objects per hectare—just above the required criterion. The negative binomial GLMM revealed a statistically significant positive effect of registration status on the total number of deadwood objects per hectare (p < 0.001), indicating that registered WKHs contained significantly more deadwood than deregistered sites. These results suggest that registration is associated with the retention of higher deadwood quantities. Model diagnostics, including Q–Q plots and Pearson residual histograms, supported the adequacy of model fit and confirmed that overdispersion was appropriately handled by the negative binomial structure.

Eleven of the 18 forests met the criterion for the number of old spruce trees (>250 years), with an average exceeding 10 trees per hectare (Fig 5). The negative binomial GLMM showed no significant effect of registration status on the number of old spruce trees (p = 0.61), indicating no difference between registered and deregistered forests. Model diagnostics supported the model fit: Pearson residuals closely followed the expected distribution, residuals were slightly skewed but centered confirming appropriate handling of overdispersion.

### 3.4 Harvest actions within deregistered key habitats since deregistration

Combining the database of harvesting reports with the maps of deregistered WKH shows that harvesting has been conducted within 874 hectares of deregistered WKH and additionally 301 hectares were reported for harvesting after deregistration (Fig 6).

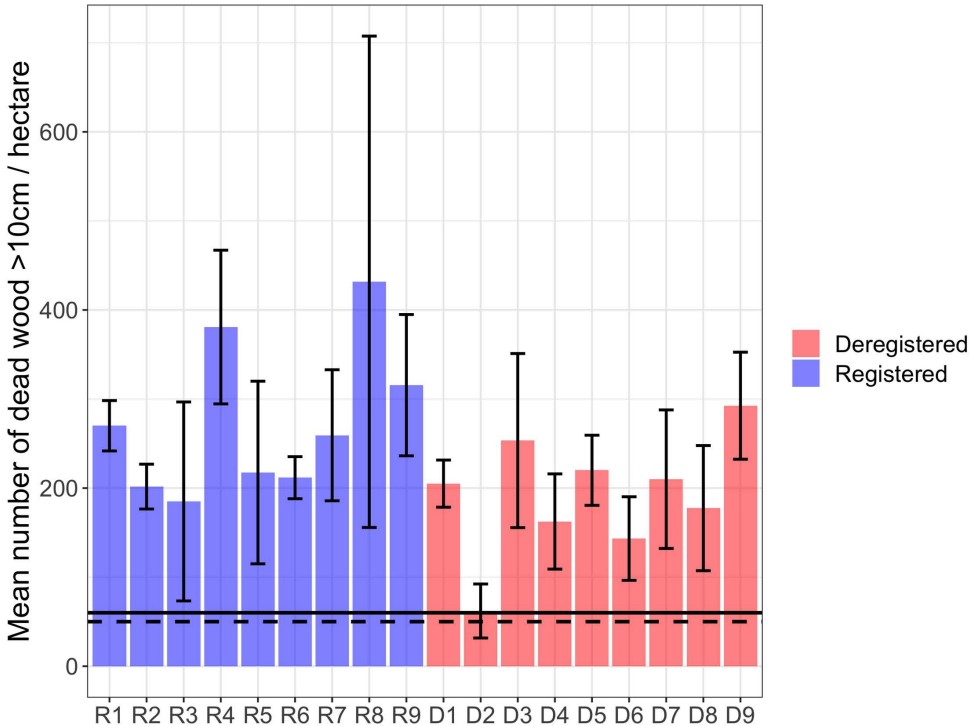

**Fig 4. Average number of deadwood objects.** Mean number (± 1 SD) for both standing and lying deadwood objects per hectare. With registered showed in purple and deregistered showed in red. The dead wood criteria are 50/hectare in Track 1 (dashed line) and 60/hectare in Track 2 (solid line).

## 4. Discussion

Our findings show that all 18 surveyed forests—both registered and deregistered—met the new, more stringent criteria and thus there were no differences in WKH status between these two groups based on the criteria in the new method. All surveyed areas represent remnants of old-growth forest in a fragmented forest landscape, typically surrounded by mires, even-aged managed forests, and clear-cuts. Despite differences in current registration status, all surveyed forests displayed core characteristics of WKH, such as the number of deadwood objects, conservation trees, trees of high age, and species of high conservation value. Signal and red-listed species were present in all forests, with six forests containing species classified as vulnerable (VU). While one deregistered WKH has been reclassified as production forest, most deregistered forests within this study are now included in a permanent protection plan for valuable mountain forests [28]. However, this plan excludes the majority of deregistered WKHs across Sweden. Moreover, although there can be various reasons behind a deregistration, such as another form of nature conservation or to register the area as a production forest, approximately 2% of the formerly registered WKH area was clear-cut or reported for clear-cutting within seven years after deregistration. Hence, the deregistration of WKH, and effects thereof, poses a threat to the conservation of valuable forest and forest biodiversity in northwestern Sweden.

### 4.1 Comparison of registered and deregistered WKH using the new classification method

This study aimed to assess the differences within registered and deregistered WKH based on the new inventory method for WKH from SFA. Our findings demonstrate that both registered and deregistered WKH consistently met or exeeded the criteria outlined in the new method. In many cases observed values for key indicators such as conservation trees, deadwood objects and trees of old age were far above the established thresholds. This suggest that during earlier WKH

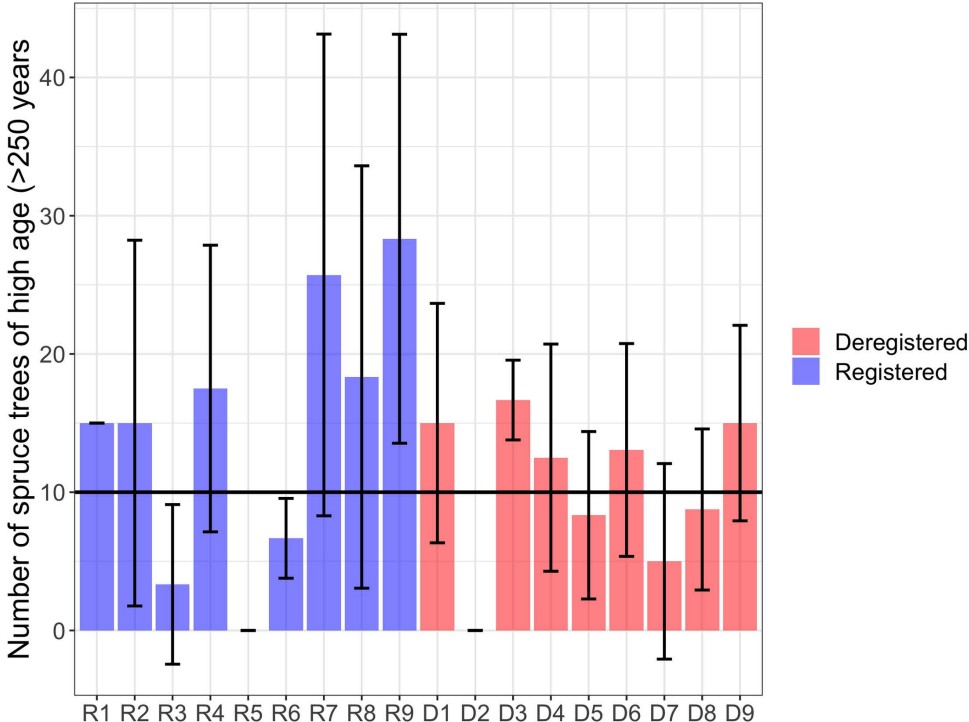

**Fig 5. Number of living spruce trees of high age (> 250 years).** Mean number (± 1 SD) of living spruce trees of high age > 250 years per hectare and in the randomly assigned sample plot in studied forests. Criteria for Track 2 (black line) in the developed method is 10 trees per hectare.

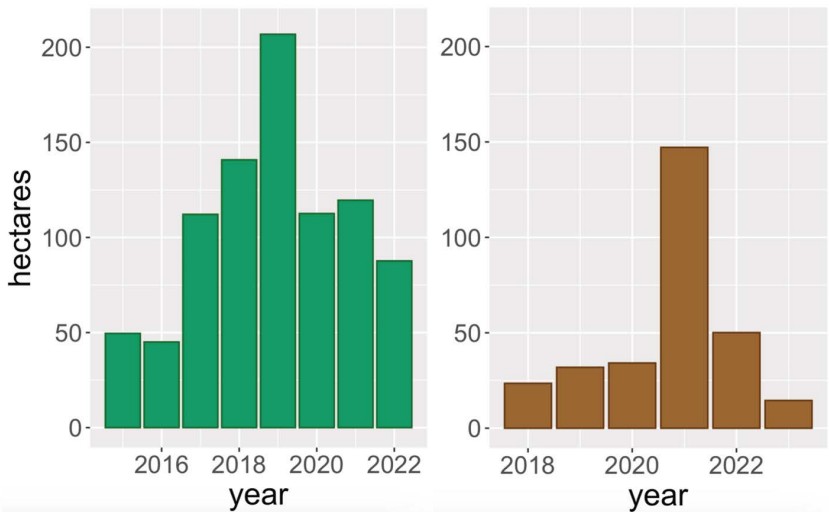

**Fig 6. Harvested forest and forest reported for cutting per year.** Harvested forest area (left) and forest area reported for harvest (right) within the deregistered key habitats in Sweden.

identification efforts, stricter criteria might already have been applied in these areas. Some differences were observed between the groups: registered WKH had slightly more conservation trees and significantly more deadwood objects. In particular, deregistered site D2 had low deadwood density, possibly influenced by past wildfire as signs of this was observed in field. Since fire can both increase or decrease deadwood volumes depending on burn severity and frequency [37,38], this highlights the importance of considering site history when evaluating forest conservation value. Despite these differences, all areas qualified as WKH under the new method, indicating deregistration was not driven by low conservation values. This underscores the need for transparent WKH evaluations and reinforces the importance of maintaining protections for deregistered sites that still retain high conservation value.

## 4.2 Effects of deregistration on future conservation values and biodiversity in boreal forests

Our findings highlight the critical need for accurate and ecologically grounded classification of WKH, as deregistration may lead to the loss of forest areas that still hold high conservation value. Despite the implementation of a more stringent assessment method, no systematic differences in overall conservation value were found between registered and deregistered WKH. This suggests that deregistration has not selectively removed lower-value sites. On the contrary, valuable forest areas remain vulnerable to deregistration and subsequent harvesting.

This is particularly concerning given that 2% of deregistered WKH in our study were already clear-cut or reported for clear-cutting. While this percentage may appear small, it represents an irreversible loss of old-growth features such as old trees, large deadwood structures, and habitat-specialist species. At least 19% of all clear-cuts between 2003 and 2019 in Sweden occurred in old-growth forests [3], reinforcing the broader pattern. The example of site D2 illustrates the consequences of deregistration: a forest with documented conservation value and a vital role in green infrastructure was made eligible for clear-cutting with only six weeks' prior notice, in line with Swedish forestry law [39]. This reflects how administrative decisions, rather than ecological degradation, can open the door to habitat loss. If deregistration continues without rigorous ecological justification, Sweden risks accelerating the loss of high-biodiversity forests. The result may be increased fragmentation, diminished ecological connectivity, and reduced resilience in boreal forest ecosystems—trends that are already recognized as key drivers of biodiversity decline and ecosystem degradation globally [40,41].

## 4.3 Current classification of deregistered WKH in Västerbotten county

The case of Abmoberget illustrates the vulnerability of intact forests with high conservation value that lack formal protection. Although parts of the deregistered WKH in our study have recently received long-term protection through a national agreement between the Swedish Environmental Protection Agency and Sveaskog [28], this coverage remains partial. This agreement protects nearly 100,000 hectares of mountain forests but of the 61,000 hectares of deregistered key habitats analyzed in this study, only 2,100 hectares were included in this agreement [28]. This indicates that many areas, such as the deregistered WKH D2, may have been reclassified as production forest despite high conservation values [42,43]. A trend supported by previous findings that at least 19% of clear-cuts between 2003 and 2019 occurred in old-growth forests [3]. It is therefore critical to further monitor the fate of all deregistered WKH, to assess the full impact of this action for future forest conservation values and biodiversity in this region.

## 4.4 Strengths and limitations of the new inventory method and the WKH concept

The revised inventory method demonstrates improved structure and aims to enhance transparency, objectivity, and predictability. It builds on three key categories used in WKH inventories across Europe: stand-level structural features, presence of individual habitat elements, and presence of indicator species [44]. However, reliance on expert judgment—especially for identifying red-listed and signal species—introduces a degree of subjectivity. Independent verification could help mitigate this risk. Although tailored for northwestern Sweden, the method does not explicitly address practical challenges such as snow cover. In our study, inventories were conducted during snow-free periods, which is essential for

reliably identifying ground-level features and species. However, under current regulations, the six-week notification period for clear-cutting may result in inventories being conducted while snow still covers key forest floor indicators required by the inventory method. Extending the notification period to ensure at least six snow-free weeks—or explicitly stating that inventories requires snow free conditions could —significantly improve accuracy in northern regions. According to our study, the new, stricter method was still able to classify WKH as before, but even stricter criteria risk excluding valuable old-growth stands in regions with many yet-unregistered WKHs, potentially undermining conservation efforts [11].

### 4.5 The role of Swedish woodland key habitats in a global perspective

By the year 2030, the EU biodiversity strategy sets out a commitment to protect at least 30% of areas of the land, to enhance biodiversity and ecosystem functions [45]. At least one third should be under strict protection and include all remaining primary and old-growth forests [45]. The focus on high yield wood production over a long period has resulted in loss of natural and diverse forests in many European regions, while the forests of north western Sweden still have areas of unique old-growth forest left [5]. Deregistration of WKH in the northwestern part of Sweden, were the highest proportion of old-growth forests is found, therefore does not align with national and international goals for biodiversity conservation and nature protection. Instead, proactive measures and a strict system of controls before harvests are essential to ensure that forests with significant nature conservation values are not replaced by clear-cuts and planted, even-aged production forests of low biodiversity.

## 5. Conclusion

We show that the new method of KWH classification likely will find similar conservation values as the old method and, thus, results in the same classification of valuable forests in northwestern Sweden. However, the deregistration of KWH is in itself concerning, as deregistered forests seem to contain the same high values as still registered forests, at the same time as they are at considerable risk of losing their values to clear-cut forests. Moreover, as emphasized by the SFA, there are most likely many still unidentified WKH and with higher criteria some of these might not be classified as WKH. These areas must be identified and protected to ensure that they can persist as refuges for threatened species in the future and act as biodiversity hotspots in the larger continuous green infrastructure. The green belt of old-growth forests in northwestern Sweden, including known and unknown, as well as registered and deregistered, key habitats, still has the potential to support high biodiversity and should therefore be excluded from forestry to meet the European biodiversity strategy.

## Supporting information

**S1 Table. Coordinates of plots.** Table S1 Shows a list of all forest areas with number of sample sites and coordinates for all sample sites. Forest areas are named R1- R9 for the registered WKH and D1 – D9 for the unregistered WKH. Sample plots within each area are labeled plot1 – plot x, depending on the number of plots.
(XLSX)

**S2 Table. Results from field survey.** Table S2 shows the full result from the field survey. Including all parameters found at each sample site within all forest areas.
(XLSX)

## Acknowledgments

We thank Gudrun Norstedt, Frank Götmark and Tero Härkönen for assisting with design of the field study as well as useful comments on the manuscript. Jonas Nordenström is thanked for helping out with species identification and support in the field. Also thanks to Swedish Society for Nature Conservation and the local forest group of Kronoberg for the work behind the map of deregistered key habitats.

## Author contributions

**Conceptualization:** ulrika Ervander.

**Data curation:** ulrika Ervander, Willian T. A. F. Silva.

**Investigation:** ulrika Ervander.

**Software:** Willian T. A. F. Silva.

**Supervision:** Karin C. Hårding.

**Visualization:** Willian T. A. F. Silva.

**Writing – original draft:** ulrika Ervander.

**Writing – review & editing:** Willian T. A. F. Silva, Karin C. Hårding, Micael Jonsson.

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
