## [Decision Letter · Decision Letter 0]

8 Aug 2024

Dear Dr. Ervander,

Thank you for submitting your manuscript to PLOS ONE. After careful consideration, we feel that it has merit but does not fully meet PLOS ONE’s publication criteria as it currently stands. Therefore, we invite you to submit a revised version of the manuscript that addresses the points raised during the review process.

We look forward to receiving your revised manuscript.

Kind regards,

Marcela Pagano, Ph.D, M.D.

Academic Editor

PLOS ONE

3. We note that Figure 1 in your submission contain copyrighted images. All PLOS content is published under the Creative Commons Attribution License (CC BY 4.0), which means that the manuscript, images, and Supporting Information files will be freely available online, and any third party is permitted to access, download, copy, distribute, and use these materials in any way, even commercially, with proper attribution. For more information, see our copyright guidelines: http://journals.plos.org/plosone/s/licenses-and-copyright.

4. We notice that your supplementary table is included in the manuscript file. Please remove them and upload them with the file type 'Supporting Information'. Please ensure that each Supporting Information file has a legend listed in the manuscript after the references list.

Reviewers' comments:

Reviewer's Responses to Questions

**Comments to the Author**

1. Is the manuscript technically sound, and do the data support the conclusions?

Reviewer #1: No

Reviewer #2: Yes

2. Has the statistical analysis been performed appropriately and rigorously?

Reviewer #1: No

Reviewer #2: Yes

3. Have the authors made all data underlying the findings in their manuscript fully available?

Reviewer #1: No

Reviewer #2: Yes

4. Is the manuscript presented in an intelligible fashion and written in standard English?

Reviewer #1: No

Reviewer #2: Yes

Reviewer #1: Review statement

PlosONE-D24-09128

Status and future prospect of deregistered woodland key habitats in northwestern Sweden

The manuscript describes the important problem of Sweden de-registering the woodland key habitats and delves into the differences between WKHs and deregitered WKHs.

The project aimed to “investigate the performance of the developed survey methods” and “compare the conservation value of registered WKHs and recently deregistered WKHs”. The introduction of study is promising and I have clearly least comments on the introduction, but unfortunately falls apart in the methods and results.

Methods section needs to be completely rewritten, it does not give the information required for a scientific article. The results section is very discussive and must be rewritten in a scientific manner, providing results of the tests, references to tables and figures that support the results. The figures are not relevant in all cases and tests results are not shown. The discussion does not set the study in a larger context. Below I give some detailed comments for improvement of the manuscript.

Detailed comments

L.33. Do you mean that primary boreal forests are located in northwestern Sweden and northwestern Finland? Later in the Introduction L. 67 you say northwestern Russia and Fennoscandia. This is unclear, please specify.

Keywords: Do not use the same words in title and keywords. The purpose of keywords is to improve findability of the article. The keywords should then be different from the title. I don’t think “deregistered woodland key habitats” is a good keyword.

L. 87 WKHs may serve… and may host.

L. 91. Define SFA the first time you use the abbreviation. Also, change Skogsstyrelsen to SFA or Swedish Forest Agency thorough the manuscript.

L. 92. Not all WKHs are old-growth forests. This part of the text gives an impression that they are. Be careful with wording to not give a wrong impression that WKH is a synonym of old-growth forest.

L.122. “identified” has a typo.

L.123. Change the “Sveriges officiella statistik" to correct, also correct the references in the reference list to be actually found.

L. 127. Please, check and correct the sentence.

L. 129. Key habitat is supposedly same as WKH? Don’t change the terminology, the reader starts to wonder if this is the same or different.

L. 132. In scientific, formal writing, “it is” is the right form, not “it’s”. This happens also elsewhere.

L. 134. You should develop more clear study questions. What is meant by performance? What is the “Conservation value”? Number of species? “We investigate” is not good enough, you need to specify what you want to get out of the research, and make sure the reader understands why this is important. Finally, the last sentence in the introduction, stating that “The study highlights the potential consequences that a switch in inventory method... may have on biodiversity”. I am not sure if I agree, all these areas were classified as WKHs before and after the inventory methods switch, right? The fact that they were deregistered had not to do with the fact that they were less good, or did it? The forest owners were given a possibility to deregister their WKHs.

L. 152-156. Unclear what is the purpose of this information here?

L. 159-167. This section does not give enough information on how the survey was conducted. How large areas were surveyed? How were they surveyed? How many plots per area were surveyed. The latitudes, longitudes, vegetationtypes, general infromation on sites should be given as well. What is old tree, what is a tree with conservation value? It is not clear enough, please give a reference that these trees have a significant importance for biodiversity if you say they have it. While writing the methods, remember that the reader must be able to repeat the study after reading what you did. CHeck your peers before submission, if you are not sure.

L. 169-177. I would need to see the protocol that is used. Alternatively, list the criteria in a table.

L. 180-189. Suddenly you say that you are using conservation trees as your response model. Why them? Why do you think they are important? You do not mention anything in your study questions about conservation trees, but the differences between WKHs and deregistered forests. In line 183 you say habitat is used as independent variable, what is habitat? Why Chi-square test?

What is the Geographical analysis? Why was it done? I see no purpose in study questions. Give the versions and references to R, state the packages used and give the references to them and QGIS. Did you check if the data fulfilled the requirements for the glms? How did you do it?

Results

The results are not written correctly. I need to see the results, not to read your statement that the results show something that I don’t see. I was wondering if this is a Results + discussion in one section, but that was not the case. The whole results section must be re-written, below I give examples of what should be corrected.

Where are the references to tables and figs showing the results of GLM and chi-square? What does 50/50 rule has to do with your results? Where is Table S3? Table S2 doesn’t exist at all in the text. “Non-replacable promise” is an odd term, I suggest you check with the forest company how they would translate it. I don’t know what this has to do with results? Doesn’t it belong to methods? You knew it already before the study, it was included in your design.

L.224. Where did the human disturbance come from? Never mentioned before. Where is the test result for the redlisted species? You say that it was a subsampling, but now what is a subsampling? Can one trust the data? (True zeros?)

L 244. You should not repeat the methods in the results. The long text 244-253 gives one result: The number of conservation value tree in WKHs and deregistered WKHs did not differ from each other (X2=??, p<0.08).

L 276-277. You should say how high was the number of deadwood and if it differs between WKHs and deregistered WKHs. I don’t understand where the whiskers come from in fig 4? What is N in this? Not clear in methods how deadwood is counted.

L. 298: This is not mentioned in the study aims, questions or methods. If this was an aim, it must be stated before the result.

Fig 6. I don’t understand the difference in left and right. Is this worth a figure?

Fig 7. Isn’t this just 2 numbers? Not worth a figure.

Discussion

L. 318-327. You give an illusion that WKHs are strictly protected, when you specifically say that non-replaceble promises” are not strictly protected. They are as good as WKHs, in general (classified most probably as NO, so no forestry operations allowed). Does the protection has to do with you study? I don’t think so. You should discuss your test results, now what happened to your sites after the study.

I didn’t read the rest of the discussion, because it did not have any references to other studies, meaning that it is not a real discussion in terms of setting your results into a larger context. You should rewrite the discussion from the beginning.

Reviewer #2: This study presents interesting results concerning the evaluation of the process of assessment of Woodland Key Habitats (WKH) in the boreal Sweden. This field work helps to evaluate the effectiveness of the criteria currently applied to evaluate primary / old-growth forests which are likely to be object of certification in Sweden and should ideally guide conservation efforts and more generally forest management planning. Specifically, the authors contrast a new set of stringent criteria/thresholds for WKH classification currently applied in the Swedish Northwest with the original set of criteria/thresholds originally widely applied to classify WKHs in Sweden. As the new criteria have caused the deregistration of some WKHs, which enables ordinary timber harvesting, the authors evaluate the possible consequences for biodiversity loss.

While the topic and the field work are certainly of interest for PLOS readers, the manuscript needs major revision to be acceptable for publication. Even if the authors focus mostly on the field sampling they did in the area, they miss to report on a more extensive vision over the process of deregistration in northwestern Sweden. Furthermore, a better description and comparison of the biological value (in terms of species or taxa) of the registered and deregistered WKHs is lacking and must be better discussed, for example commenting on the level of similarity between deregistered and registered WKHs concerning signal and red-listed species or on the IUCN red-list status.

The paper is difficult to read and often it reports concepts more than once without going deep in their explanation. Specifications on the details of the results lack generality and do not easily deliver messages that can be exported.

The statistical methods must be better explained and their application in the results must be clearly specified.

Finally, even though WKHs are part of a classifying scheme of the forest and do not have legal value for forest protection, in my opinion, the authors must do an effort to envisage their potential role to meet the 2030 targets of the EU Nature Restoration law by adding raw calculations in this sense. This would much increase the value of the paper.

Detailed comments are exposed in the attached pdf.

**Do you want your identity to be public for this peer review?** For information about this choice, including consent withdrawal, please see our Privacy Policy

Reviewer #1: No

Reviewer #2: No

---

## [Author Response · Author response to Decision Letter 1]

13 Jan 2025

Answers to Journal requirements:

A: The code used is ready to be shared and will be made avaliable upon publication of the work

3. We note that Figure 1 in your submission contain copyrighted images. All PLOS content is published under the Creative Commons Attribution License (CC BY 4.0), which means that the manuscript, images, and Supporting Information files will be freely available online, and any third party is permitted to access, download, copy, distribute, and use these materials in any way, even commercially, with proper attribution. For more information, see our copyright guidelines: http://journals.plos.org/plosone/s/licenses-and-copyright.

A: The correct copyrite information has been added to the map and ledgend for Fig 1.

4. We notice that your supplementary table is included in the manuscript file. Please remove them and upload them with the file type 'Supporting Information'. Please ensure that each Supporting Information file has a legend listed in the manuscript after the references list.

A: The supplementary table is now removed and added to the manuscript as Table 1 since there were a lot of comments on both the method and the results which we hope can be more clear when showing the summary from the field survey in the the text.

---

## [Decision Letter · Decision Letter 1]

23 Apr 2025

Dear Dr. Ervander,

Thank you for submitting your manuscript to PLOS ONE. After careful consideration, we feel that it has merit but does not fully meet PLOS ONE’s publication criteria as it currently stands. Therefore, we invite you to submit a revised version of the manuscript that addresses the points raised during the review process.

Dear authors, please, see reviewer comments to improve your manuscript,

We look forward to receiving your revised manuscript.

Kind regards,

Marcela Pagano, Ph.D, M.D.

Academic Editor

PLOS ONE

Journal Requirements:

Reviewers' comments:

Reviewer's Responses to Questions

**Comments to the Author**

Reviewer #1: (No Response)

Reviewer #2: All comments have been addressed

2. Is the manuscript technically sound, and do the data support the conclusions?

Reviewer #1: Partly

Reviewer #2: Partly

3. Has the statistical analysis been performed appropriately and rigorously?

Reviewer #1: I Don't Know

Reviewer #2: Yes

4. Have the authors made all data underlying the findings in their manuscript fully available?

Reviewer #1: Yes

Reviewer #2: Yes

5. Is the manuscript presented in an intelligible fashion and written in standard English?

Reviewer #1: No

Reviewer #2: No

Reviewer #1: The manuscript has been improved a little, but many of my previous issues are not solved. The study question is still very vague “Investigate the performance of a survey methods and compare the conservation value of registered and deregistered WHKs.” I commented this already during the first revision, stating “You should develop more clear study questions. What is meant by performance? What is the “Conservation value”? Number of species? “We investigate” is not good enough, you need to specify what you want to get out of the research, and make sure the reader understands why this is important.” In the revised manuscript, instead of developing more clear questions, you have added “In this study…” (L. 143 in the tracked-changes-document). A clear question would be for example: ”We asked whether de-registered WKHs have lower conservation value in terms of the 1) number of conservation trees, 2) redlisted species and xxx”. I don’t really understand how “performance” is investigated, as you have nothing to compare in reagrds to performance.

The methods section has been improved slightly, but still I don’t know how the inventory was done. How large are the sample plots? I checked the supplementary files, and found them to be rather bad, partly in Swedish and partly English, and not understandable (Table S1: NB1, NB2…, Table S2: KH1, KH2,…, in the figure 2. R1,R2, …,D9. The addition in Line 185-187 does not help at all.

You mention in your reply that you have corrected “habitat” to “WKH” in the methods section 2.3., but it still says habitat and I was again puzzled. “Habitat” is your individual stand, right? Then it would not help to change it to “WKH”, I would be as puzzled. WKH ID or Forest stand ID could work instead.

Later you say (in the tracked changes version L 210—216) that you fitted glm with registration status and habitat as independent variables. But you had only one observation of redlisted species per habitat/stand, right? And then you used chi-square test? Why? I really don’t follow, and you did not provide answers in your response letter, even though this question was asked already in the first round. You have also deleted important information from the text, the Poisson error distribution. Or didn't you finally use Poisson? Also, it would be important to check the model fit in all cases. You had changed from glmm to linear mixed model, why? Number of trees, number of species or number of deadwood pieces as a response variable requires Poisson error distribution, most probably, which makes lmm an incorrect method. At this point, I don’t know what kind of analyses you have finally conducted, it is so unclear that I totally lost it.

I said earlier that your R and packages must be cited. Version is not enough. This remains an issue.

The first part of the results is still very discussive, not much has changed. You mention that you have added a lot of text of statistical analyses in the results, but I see barely any, some p-values and Variance values are added. No real results are shown for the models, stating just p-vales is not sufficient in reporting the results. Boxplots are not a correct visualization method when n<5 (which I see is true in many of the forest stands (Supplementary tables 1 and 2)). I did not realize this low n earlier.

The sentence in the results: “These findings suggest that registration status significantly influences deadwood levels, with habitat-specific variation also playing a role.”, is not true. How would the registration status affect the levels of deadwood? If the status is changed from registered to de-registered, would the deadwood amount suddenly decrease? This is a common wording failure, be careful! Further, the sentence would belong to the discussion, not results.

Discussion is still vague. There are many methods to evaluate conservation values of forests in Europe, these are not mentioned and the results are not discussed with the light of those findings. Most of the articles cited in the manuscript are Swedish reports by SFA, which shows that the manuscript may not reach wider, international audience. The discussion is not fully supported by the results, for example the issue of snow cover comes very surprisingly up in the discussion, as the timing of the inventories is not addressed anywhere in the results or elsewhere. For the reader that knows Sweden and is familiar with the system, this may make sense, but not to anyone else.

Finally, the manuscript was earlier given a “major revision” and you were given lots of comments how to improve the manuscript. Unfortunately, I don’t see much improvement, many of the issues remain and the sloppiness in writing and referring to tables and figures continues. Please, next time you submit this manuscript, pay attention to actually do the changes and not just comment that you have done them. Make sure to using spelling check to correct all the typos, there are lots of typos in the text which gives a sloppy impression of the author(s). In the discussion, there are sections where nearly every sentence has a typo.

Detailed comments:

-Introduction claims that Sweden would be 4000 km long, which is not true.

-Make sure to write scientific names with capitalized Genus, and using italics (Table 1).

-You still have “human disturbance” as your result without ever mentioning it in the introduction or methods or showing any results regarding this. I was as puzzled this time as well. If you present a result of visible signs of human disturbance, it must be stated in the methods.

-In the text, after Figure 1 comes Figure 4, then 2. Please make sure all the tables and figures are correctly numbered and referred. They are not.

Reviewer #2: The paper has improved from the previous version but still needs major revision and strong review of the way results are expressed and summarized. Statistical tests must be better reported. English language must be improved. The topic of the paper is important and deserves more efforts from tehe authors to reach a publication level.

**Do you want your identity to be public for this peer review?** For information about this choice, including consent withdrawal, please see our Privacy Policy

Reviewer #1: No

Reviewer #2: No

---

## [Author Response · Author response to Decision Letter 2]

7 Jul 2025

Dear Editor,

All comments from both Editor and Reviewers has been adressed in the Response to reviewers file added to the submission.

For the requirements from the journal, the reference list has been thoroughly reviewed, and changes have been made to ensure accuracy and completeness.

Kind regards

Ulrika Ervander

---

## [Editor Report · Decision Letter 2]

13 Aug 2025

Status and future prospect of deregistered woodland key habitats in 

northwestern Sweden

PONE-D-24-09128R2

Dear Dr.ulrika Ervander,

We’re pleased to inform you that your manuscript has been judged scientifically suitable for publication and will be formally accepted for publication once it meets all outstanding technical requirements.

Kind regards,

Marcela Pagano, Ph.D, M.D.

Academic Editor

PLOS ONE

Additional Editor Comments (optional):

Please, check all the authors of the manuscript. 
---

## [Editor Report · Acceptance letter]

PONE-D-24-09128R2

PLOS ONE

Dear Dr. Ervander,

I'm pleased to inform you that your manuscript has been deemed suitable for publication in PLOS ONE. Congratulations! Your manuscript is now being handed over to our production team.

Kind regards,

on behalf of

Dr. Marcela Pagano

Academic Editor

PLOS ONE